organic chemistry

Pinnick oxidation, density functional theory simulations, transition state, molecular dynamics, molecular orbital theory, oxidation

**Author for correspondence:**
Aqeel A. Hussein
e-mail: aahh1f19@soton.ac.uk

This article has been edited by the Royal Society of Chemistry, including the commissioning, peer review process and editorial aspects up to the point of acceptance.

# Mechanistic investigations on Pinnick oxidation: a density functional theory study

Aqeel A. Hussein[1,2], Azzam A. M. Al-Hadedi[3], Alaa J. Mahrath[4], Gamal A. I. Moustafa[2,5], Faisal A. Almalki[6], Alaa Alqahtani[6], Sergey Shityakov[7] and Moaed E. Algazally[1]

[1]Faculty of Dentistry, University of Al-Ameed, Karbala PO Box 198, Iraq
[2]Department of Chemistry, University of Southampton, Southampton, Hampshire SO17 1BJ, UK
[3]Department of Chemistry, Faculty of Science, University of Mosul, Mosul, Iraq
[4]Department of Chemistry and Biochemistry, College of Medicine, University of Babylon, Babylon, Iraq
[5]Department of Medicinal Chemistry, Faculty of Pharmacy, Minia University, Minia 61519, Egypt
[6]Department of Pharmaceutical Chemistry, Faculty of Pharmacy, Umm Al-Qura University, Makkah 21955, Saudi Arabia
[7]Department of Anesthesia and Critical Care, University of Würzburg, 97080 Würzburg, Germany

AAH, 0000-0002-9259-9609; FAA, 0000-0003-4048-1526

A computational study on Pinnick oxidation of aldehydes into carboxylic acids using density functional theory (DFT) calculations has been evaluated with the (SMD)-M06-2X/aug-pVDZ level of theory, leading to an important understanding of the reaction mechanism that agrees with the experimental observations and explaining the substantial role of acid in driving the reaction. The DFT results elucidated that the first reaction step (FRS) proceeds in a manner where chlorous acid reacts with the aldehyde group through a distorted six-membered ring transition state to give a hydroxyallyl chlorite intermediate that undergoes a pericyclic fragmentation to release the carboxylic acid as a second reaction step (SRS). [1]H NMR experiments and simulations showed that hydrogen bonding between carbonyl and *t*-butanol is unlikely to occur. Additionally, it was found that the FRS is a rate-determining and thermoneutral step, whereas SRS is highly exergonic with a low energetic barrier due to the $Cl(III) \rightarrow Cl(II)$ reduction. Frontier molecular orbital analysis, intrinsic reaction coordinate, molecular dynamics and distortion/interaction analysis further supported the proposed mechanism.

# ROYAL SOCIETY OF CHEMISTRY

**Figure 1.** (*a*) Pinnick oxidation and (*b*) its well-known mechanism.

# 1. Introduction

The oxidation of aldehydes to carboxylic acids is a fundamental transformation in organic chemistry that is applied in many syntheses (examples of the use of Pinnick oxidation in synthesis of some bioactive molecules [1–5], Pinnick-type protocol for amidation of aldehyde [6]). Despite the variety of reagents available to achieve this oxidation (for general transformation of aldehyde into carboxylic acid, see [7–21]), few of them are suitable for a broad range of aldehydes and/or tolerate a wide range of functional groups [22,23]. Beside selectivity problems, high costs and complexity of the reaction conditions made the application of this transformation on a large scale reasonably difficult. By contrast, these drawbacks decrease with Pinnick oxidation. Pinnick oxidation has been widely used in organic synthesis due to the flexibility with many sensitive functionalities and suitability for sterically hindered aldehydes [22,23] (for development of Pinnick oxidation, see [24–34]). Although this reaction was first developed by Lindgren & Nilsson [27] and then by Kraus [26,28] modifications developed by Pinnick were later demonstrated to be an efficient approach to oxidize α,β-unsaturated aldehydes to their corresponding acids by using sodium chlorite under mild conditions (figure 1*a*) [31]. The chlorite protocol has been extensively applied to aliphatic, aromatic and hetero aromatic aldehydes [22,35]. The reaction mechanism was proposed to commence with protonation of the aldehyde by chlorous acid ($HClO_2$) to give intermediate **3** followed by an attack of the chlorite ion to carbonyl group to form hydroxyallyl chlorite intermediate **4** and the latter undergoes a pericyclic fragmentation to release carboxylic acid **2** and hypochlorous acid (HOCl) as a by-product (figure 1*b*) [27]. Moreover, it was found that HOCl can cause an undesired side reaction and destroy the $NaClO_2$ reactant [27,35,36]. Thus, HOCl scavengers such as 2-methyl-2-butene, resorcinol, sulfamic acid and hydrogen peroxide are added to increase the efficiency of the reaction [23,27–29,37].

Interestingly, the reaction mechanism of Pinnick oxidation is still not fully understood although the currently proposed one is acceptable, and overall this basic proposal has important considerations to be taken into account. Herein we interpret, for the first time, a DFT study on the mechanistic pathway of classical Pinnick oxidation, highlighting a detailed description of this transformation with the support of experimental [1]H NMR (figure 2). In this regard, molecular dynamics (MD) simulations were also implemented to assess the dynamics of the transition state (TS) at the atomic level. In addition, a distortion/interaction model was used to relate the activation energy of the first reaction step (FRS) to the deformation energy required to achieve the transition structure and understand the favourable interactions between the two distorted reactants that form TS.

**Figure 2.** Study summary for the Pinnick oxidation. EDG and EWG are electron-donating and electron-withdrawing groups, respectively.

## 2. Computational details

All quantum calculations were performed using Gaussian 09 program [38], where all geometries were fully optimized at the functional M06-2X [39,40] with the basis set aug-cc-pVDZ [41–43]. All minima intermediates were verified by the absence of negative eigenvalues in the vibrational frequency analysis except transition state structures being visualized by animating the negative eigenvector coordinate. All TSs were located using the Berny algorithm [44,45]. The thermal corrections were evaluated from the unscaled vibrational frequencies to obtain the free energies at 298.15 K. The effect of solvent was fully included in the optimization procedure via the solvation model based on density (IEFPCM-SMD) using $t$-butanol ($t$-BuOH) as a representative solvent medium [46]. To find the minimum energy path on the potential energy surface (PES), an intrinsic reaction coordinate (IRC) calculation in the presence of $t$-BuOH was performed for the identified TS using the Hessian-based predictor-corrector integrator, in which the Hessian was recomputed every three predictor steps with a step size of 0.05 Bohr [47–49]. All activation free energies are quoted relative to infinitely separated reagents. Classical molecular dynamics trajectory calculation, using the atom-centred density matrix propagation (ADMP), on TS was initialized in the region of the PES near to the TS and performed using Gaussian 09 in the presence of implicit solvation model (SMD/$t$-BuOH) and under standard conditions ($T = 298.15$ K and 1 atm) [50,51]. The TS was initiated into forward and backward propagations showing the product and reactants (aldehyde and chlorous acid), in which a time step of 1.0 fs was used over periods of 300.0 fs. A default set for density kinetic energy and fictitious electron mass were applied with a fully converged self-consistent field (SCF) without reporting the band gap at each step. All structures are illustrated using CYLview [52].

## 3. Results and discussion

Generally, our DFT calculations on Pinnick oxidation were carried out on a simple $\alpha,\beta$-unsaturated aldehyde, acrylaldehyde (**7**, $R^1 = H$), to evaluate the general mechanism, and on substituted cinnamaldehyde to understand the effect of electronic groups. The oxidation of aldehyde to carboxylic acid was considered to occur on two steps; FRS represents the addition of chlorite ion to carbonyl moiety with or without acid additives, whereas second reaction step (SRS) depicts the pericyclic fragmentation to release the carboxylic acid and hypochlorous acid (ClOH).

Initially, considering the reaction without acid, a TS for the reaction of sodium chlorite (NaClO$_2$) with the aldehyde group was not located. We directly turned into the fact that in the presence of acid (NaH$_2$PO$_4$) formation of chlorous acid (HOClO) by the proton–sodium exchange is thermodynamically

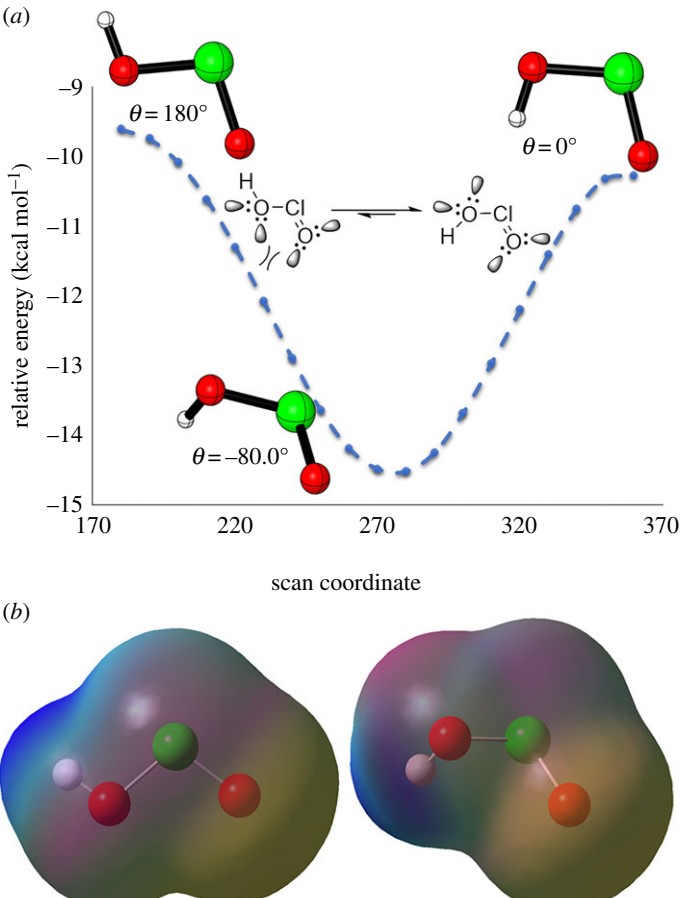

**Figure 3.** (*a*) Potential energy scan of HOClO. (*b*) Visualization of the electrostatic potential mapped with total electron density.

reasonable ($\Delta G_{r,t-\text{BuOH}} = 7.6$ kcal mol$^{-1}$ in SMD/$t$-BuOH or $\Delta G_{r,\text{H2O}} = 0.5$ kcal mol$^{-1}$ in SMD/H$_2$O) and to be correspondingly added to the aldehyde group. Reported literatures and textbooks clearly explain that the addition of HOClO is a multi-step process, as shown in figure 1*b*; where protonation of carbonyl occurs first giving oxonium ion to facilitate the addition of chlorite ion as a second step. Within our calculations, a possible TS of proton transfer from HOClO, as a separate step from the addition of chlorite ion, was not found and, therefore, any complexation between chlorous acid and carbonyl was not formed. Consequently, these results gave the indication that HOClO can be added in a different way considering different conformations of HOClO.

To probe the conformational preferences of the HOClO, we examined its potential energy scan (figure 3*a*). Calculations show that HOClO with a dihedral angle of −80° as a *cis* conformer is more stable by around 5.0 kcal mol$^{-1}$ than a *trans* conformer ($\Theta = 180°$), whereas a zero dihedral angle is slightly more stable than $\Theta = 180°$ by 0.7 kcal mol$^{-1}$. This changed our description for the addition of chlorous acid to the aldehyde group. We believe that the stability of *cis* conformation arises from the repulsion between non-bonding electrons on both oxygen atoms on HOClO (figure 3*a*). In the case of $\Theta = -80°$, the lone pairs on O = and HO moieties are far away from each other, whereas in the case of $\Theta = 180°$, these pairs are close to each other resulting in electrostatic repulsions. This belief was supported by the electrostatic potential (ESP) map for *cis* and *trans* conformations (figure 3*b*). The charge distribution of electrons on HOClO shows that the red area represents the highest ESP energy and the blue area represents the lowest ESP energy, whereas the intermediary colour is the intermediary ESPs. Finally, in the *cis* HOClO, the electrostatic repulsions are minimized, leading to favourable interaction between chlorous acid and carbonyl.

We then explored the mechanism of the reaction based on a *cis* HOClO in the presence and absence of explicit *t*-BuOH (figure 4). DFT calculations without hydrogen bonding revealed that the addition of HOClO to carbonyl occurs in a concerted manner showing a distorted six-membered TS **8** with a barrier of 20.2 kcal mol$^{-1}$ to give a thermodynamically thermoneutral hydroxyallyl chlorite intermediate **9** ($\Delta G_r = 5.2$ kcal mol$^{-1}$) (figure 4*a*). The TS **8** shows that the proton of HOClO is more likely to be near

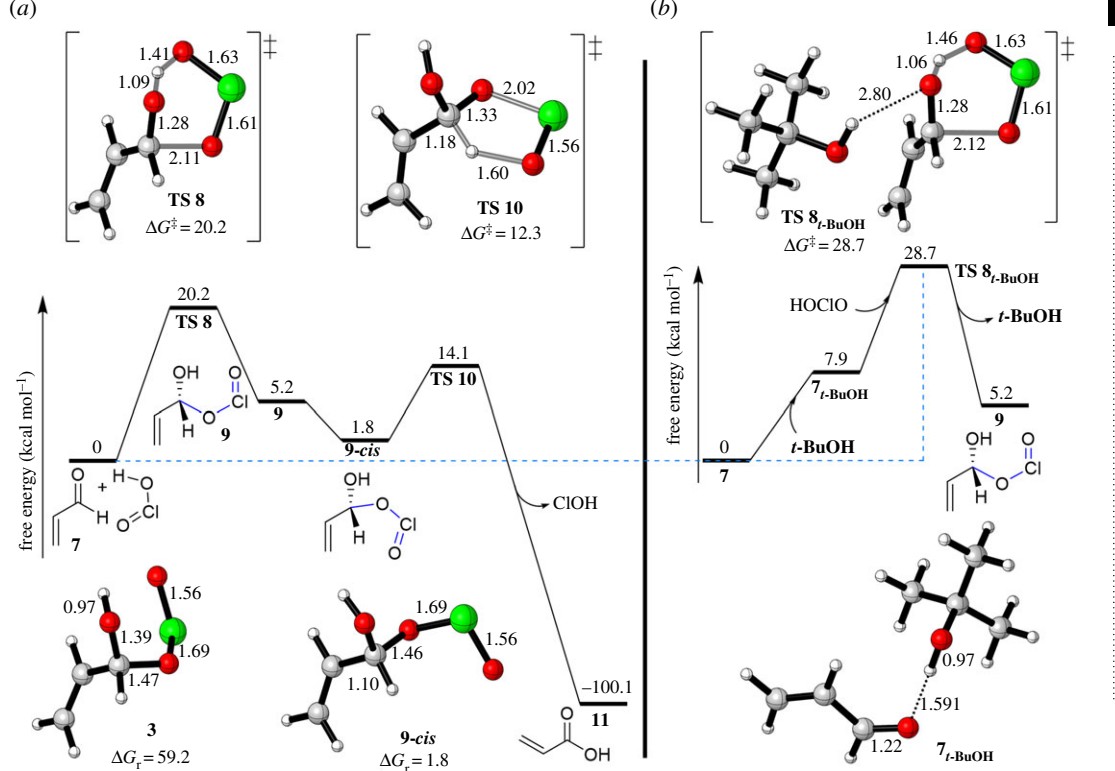

**Figure 4.** Energy profile for oxidation of acrylaldehyde (7) to acrylic acid (11); In the absence (*a*) and presence (*b*) of explicit *t*-BuOH.

the carbonyl's oxygen, establishing a shorter bond length of 1.09 Å for H–O = [C] and a longer one of 1.41 Å for HO–[Cl]. On the contrary, the O=Cl moiety was distanced at 2.11 Å from the carbon's carbonyl (C–O[Cl]). Moreover, the imaginary frequency of 445 cm$^{-1}$, referring to the vibration of bonds O–H–O[Cl] and C–O[Cl] for this TS, contributed to the similar distances for chlorine–oxygen bonds, namely: HO–[Cl] = 1.63 Å and [Cl]–O = 1.60 Å.

Turing to the effect of explicit solvent on FRS, our calculations show an excellent agreement with the experiment. Although the hydrogen bonding between the acrolein (7) and *t*-BuOH was shown to be a weak one at distance of 2.80 Å (generally, there are three different types of hydrogen bonding (H.B) categorized by donor–acceptor distances. (a) 1.2–1.5 Å as strong H.B which is mostly covalent of 40–14 kcal mol$^{-1}$: [53] (b) 1.5–2.2 Å as moderate H.B of 14–4 kcal mol$^{-1}$ which is mostly electrostatic: [54] (c) 2.2–3.2 Å as weak H.B of 4–0 kcal mol$^{-1}$ which is electrostatic: [55]), the interaction energy was found to be disfavoured ($\Delta G_{hyd} = +7.9$ kca mol$^{-1}$) (figure 4*b*). This disfavourable interaction led to an increase in the barrier of the FRSto 28.7 kcal mol$^{-1}$ (TS $8_{t-BuOH}$), thus ruling out the possible intermediacy of hydrogen bonding (via TS $8_{t-BuOH}$) due to the steric hindrance around the OH of *t*-BuOH. This result was supported by experimental $^1$H NMR study[1] (figure 5). The results showed that the OH signal has not been significantly affected by the carbonyl group although we observed a slightly deshielded shift of the OH signal in CDCl$_3$ (figure 2*c*) without being accompanied by signal broadening. Pinnick oxidation normally runs in a hugely excess amount of the solvent, so this hydrogen bonding would be out of effect, instead intermolecular hydrogen bonding between solvent molecules would rather be predominant. Thus, even with the slight shift in the OH signal, the reaction proceeds through the manner of the TS **8** progression (figure 4*a*). Moreover, Pinnick oxidation has also been conducted in aprotic solvents like acetonitrile and THF which supports the case for excluding the solvent bonding with TS **8**.

Returning to the mechanism, hydroxyallyl chlorite intermediate **9** needs a rotation to release acid via pericyclic fragmentation as SRS, where C=O=[Cl] = O moiety is *trans* to the hydrogen atom of aldehyde

$^1$H NMR spectra were recorded in CDCl$_3$ (purchased from Cambridge Isotope Laboratories, Inc.) at 298 K using Bruker AVII400 (400 MHz) spectrometer. Chemical shifts values (δ) are reported in ppm relative to residual chloroform (δ 7.27 ppm for $^1$H). All spectra were reprocessed using ACD/Labs software v. 12.1.

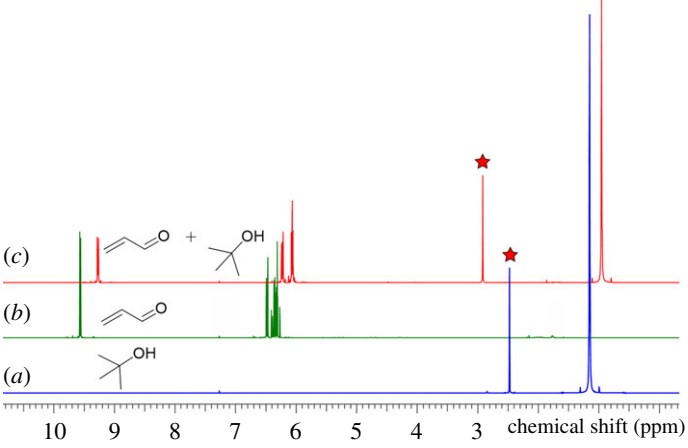

**Figure 5.** $^1$H NMR of (*a*) *t*-BuOH, (*b*) acrolein (**7**) and (*c*) 1 : 4 mixture of **7**:*t*-BuOH. Red star indicates the position of the very weak deshielded OH signal of interest.[18.]

(figure 4). So, the [Cl] = O moiety must face the hydrogen's aldehyde in a *cis* conformation **9-cis**. The intermediate **9-cis** was calculated to be more stable than *trans* by 3.4 kcal mol$^{-1}$ (see electronic supplementary material, figure SI1 for the relaxed potential energy scan for the dihedral angle of H–C–O–Cl bonds in intermediate **9** in electronic supplementary material). Releasing acids was found to be kinetically and thermodynamically accessible. DFT calculations revealed that the energy barrier for the fragmentation step is 12.2 kcal mol$^{-1}$ as a very thermodynamically favourable step ($\Delta G_r = -101.9$ kcal mol$^{-1}$). This high exergonicity represents a reasonable result because of the reduction in the oxidation state of chlorine from Cl(III) to Cl(I). It was reported that Pinnick oxidation is an exothermic reaction, in which a large scale process should be run at 0 C [5]. In comparison with SRS, the FRS is rate-determining due to the double barrier needed for the FRS.

## 3.1. FMO analysis of the FRS

The highest occupied molecular orbital (HOMO) and lowest unoccupied molecular orbital (LUMO) are shown on the aldehyde and chlorous acid (figure 6) [56,57]. Results show that HOMO-1 belongs to the HOMO of the alkene part whereas HOMO is the HOMO of carbonyl as an energetically higher orbital. On the one hand, an energy gap of 8.24 eV when aldehyde is HOMO part (HOMO = −9.20 eV) and acid is LUMO part (LUMO = −0.96 eV). On the other hand, the opposite configuration of orbital overlapping results in a slightly higher gap of 8.37 eV. That means chlorous acid is the LUMO part and carbonyl group is the HOMO part. This seems to us to be an inverse-electron demand ene-type reaction. There is a broad similarity between our findings and those for the ene-type reaction (For ene-type reaction see: [58,59]). However, different substituents, EDGs or EWGs, on the alkene site would change this orbital preference (see below). Visualization of the HOMO orbitals for TS **8** shows the orbital overlapping between the HOClO and carbon's carbonyl (figure 6).

## 3.2. IRC and MD simulations of the FRS

To have a better visualization of the FRS, IRC and MD simulations were carried out with SMD/*t*-BuOH. The IRC result of the hydroxyallyl formation, shown in figure 7, points out three important considerations. Firstly, although the reaction is not forming a cyclic product, its TS forms a distorted six-membered ring to pass the FRS due to the required orientation between the *cis* conformation of the HOClO and carbonyl groups. Secondly, proton transfer from chlorous acid represents the first-half pathway, from the pre-reaction complex until the TS, with a noticeable change of the bond length near TS. The bond length of O–HO[Cl] in the pre-TS zone is 1.45 Å whereas 1.03 Å for H–O[Cl], and once reactants reach the middle pathway, TS zone, the O–HO[Cl] bond length shortens to 1.09 Å and H–O[Cl] to 0.45 Å. After passing the TS to the product zone, the second-half pathway from the TS to the product, the bond H–O[Cl] disappears with a maximum growth of the hydroxyallyl group and at the same time, C–O[Cl] bond starts growing giving the hydroxyallyl chloride intermediate. Thirdly, the accounted barrier for FRS has shown to be totally consumed for the proton transfer and not the

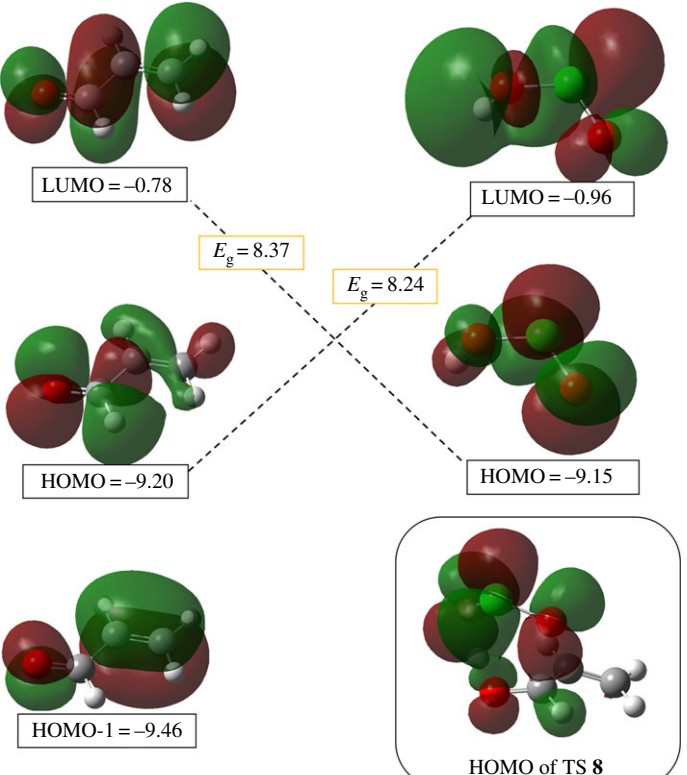

**Figure 6.** FMO analysis for the FRS. Energies are in eV.

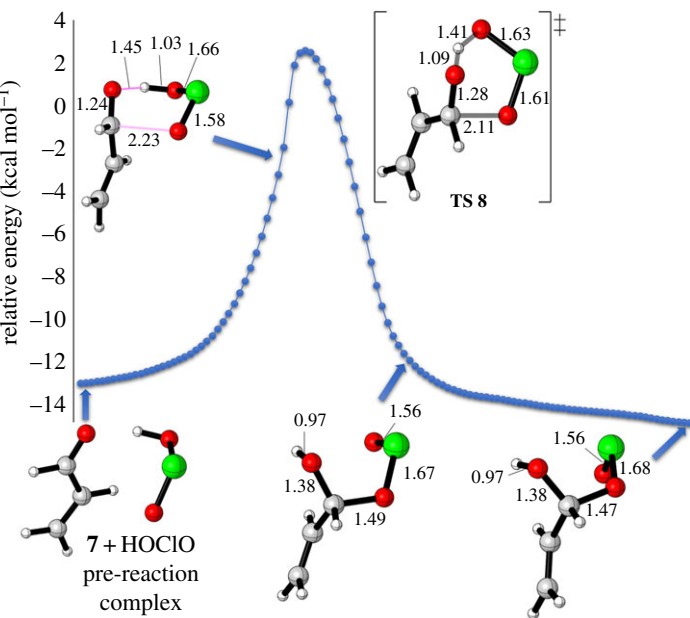

**Figure 7.** IRC of the FRS showing snapshots along the pathway.

addition of O = [Cl] moiety. However, without proton transfer, a TS for the addition of O = [Cl] to the carbonyl was impossible to locate. Further validation on IRC of the FRS was obtained from molecular dynamics simulations (see below).

Due to significant differences between the progressions of the two bond formations O–H and C–O in the TS **8**, it would be valuable to get more validations on PES towards the chronological formation of these bonds from the point of molecular dynamic (MD) simulations, especially the behaviour of the system within time (figure 8) (for the molecular dynamics on transition state

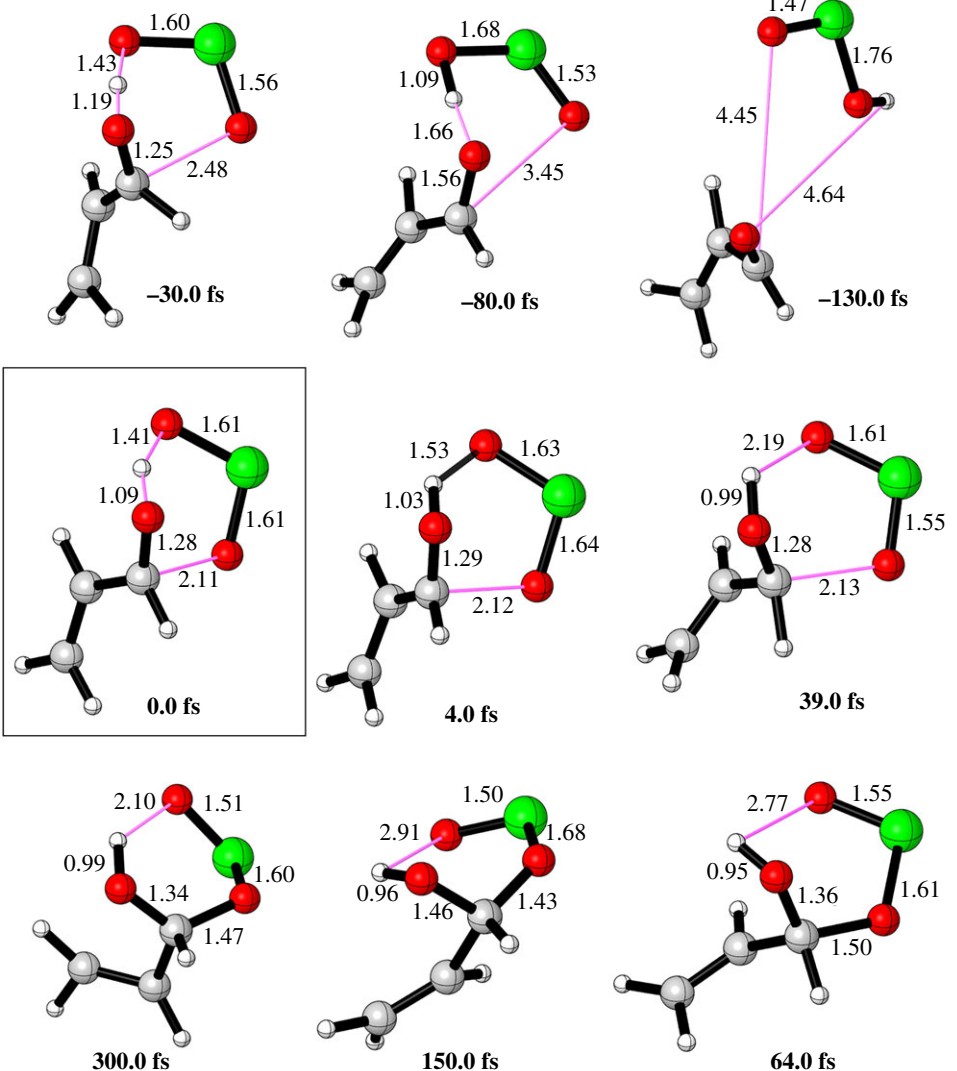

**Figure 8.** Snapshots from a single typical FRS trajectory for forward propagation leading to hydroxyalkyl chlorite formation and backward propagation leading to acrylaldehyde and chlorous acid. Trajectories initiated in the region of the potential energy surface near TS (time = 0 fs) showing reactive bonds.

structures see: [60–66]). Classical MD calculations were carried out on one trajectory, where the forward and backward propagation of TSs are initiated in the region of the PES near the TS ($t =$ 0 fs) showing the typical reactive bonds toward intermediate **9** and reactants (acrylaldehyde and chlorous acid). Here, the time between the first and second bond formations during the FRS pathway is the asynchronous time or the timing gap. Figure 7 shows snapshots for the forward and backward propagation of TS **8** at the time of the first and second bond formation. In general, a greater development of the O–HO[Cl] bond relative to the C–O[Cl] bond was noticed, which agrees with the IRC results in terms of the general outcomes (product or reactants). Although IRC results showed that most of the barrier for FRS returns to proton transfer which would suggest a long time for proton transfer, MD simulations indicate a very short time of 4.0 fs at O–H = 1.03 Å and H–O[Cl] = 1.53 Å at the product formation zone. While the C–O[Cl] bond is still in the transition state 1.03 requires a further 60.0 fs to be fully formed and leaving the transition zone into the product zone (C–O[Cl] = 1.50 Å, 64.0 Å). That means the timing gap is 60.0 fs between proton transfer and addition to oxonium ion. For the backward propagation, at 30.0 fs TS 2 starts leaving the transition zone into the reactant zone and then at 80.0 fs TS **8** enters the pre-complex reactant zone. At 130.0 fs, both the acrylaldehyde and chlorous acid were fully separated with more than 4.5 Å distance between them. Interestingly, at 130.0 fs, *trans* conformation of chlorous acid is formed and this would suggest that the timing gap between *cis* and *trans* HOClO is probably 50.0 fs.

**Table 1.** Barrier, reaction and gap energies for the effect of electronic groups on cinnamaldehyde oxidation (X-Ph-CH = CH-CHO). Free energies are in kcal mol$^{-1}$ and energy gap ($E_g$) are in eV.

| entry | X | $7 \rightarrow 9$ via TS 8 | | | $9$-*cis* $\rightarrow 11$ via TS 10 | | |
|---|---|---|---|---|---|---|---|
| | | $\Delta G^{\ddagger}$ | $\Delta G_r$ | $E_g{}^a$ | $E_g{}^b$ | $\Delta G^{\ddagger}$ | $\Delta G_r$ |
| 1 | H (7a) | 21.9 | 6.8 | 6.94 | 7.81 | 10.8 | −102.1 |
| 2 | 4-NO$_2$ (7b) | 22.1 | 5.7 | 7.41 | 6.95 | 11.8 | −100.9 |
| 3 | 4-Cl (7c) | 20.9 | 7.3 | 6.91 | 7.70 | 12.0 | −102.4 |
| 4 | 4-CH$_3$ (7d) | 20.5 | 6.0 | 6.71 | 7.86 | 11.7 | −102.8 |
| 5 | 4-OCH$_3$ (7e) | 19.8 | 6.4 | 6.38 | 7.95 | 11.9 | −102.9 |

$^a$Energy gap when aldehyde is HOMO and HOClO is LUMO.
$^b$Energy gap when HOClO is HOMO and aldehyde is LUMO.

## 3.3. Effect of electronic groups

Next, we probed the effect of electronic groups on cinnamaldehyde; EWG and EDG are tested on FRS and SRS (table 1). Geometries of TSs during the rate-limiting step (FRS) are shown in figure 9 whereas the TSs and intermediates of the SRS are shown in supplementary information. DFT calculations show reasonable trends for both EWG and EDG and generally they correlated very well based on the barriers of TS 8 which adds further validation to the level of theory being used in this system. Donating groups, entries 4 and 5, have decreased the barrier of the FRS whereas in the EWG group, entry 2, showed opposite behaviour except 4-Cl cinnamaldehyde, entry 3, whereas the unsubstituted cinnamaldehyde (entry 1, X = H) displayed a moderate barrier. In the case of EWG, the electronic deficiency increases on the oxygen's carbonyl and results in a weaker deprotonation along the six-membered ring TS due to the decrease in nucleophilicity and at the same time the electrophilicity of the carbonyl group increases. These activation barriers are consistent with the calculated energy gap. Interestingly, since FRS TS consists of deprotonation and oxygen addition, both EDG and EWG groups are more likely to be effective on the deprotonation step.

The energy gaps of the FRS obtained from Frontier molecular orbital (FMO) analysis show that HOClO is the LUMO part and aldehyde is the HOMO part except for 4-NO$_2$ cinnamaldehyde, entry 2. Moving to the SRS, the electronic groups have not shown a positive effect in which unsubstituted cinnamaldehyde has the lowest barrier of 10.8 kcal mol$^{-1}$ with a very thermodynamically favourable fragmentation step of −102.1 kcal mol$^{-1}$. EDGs and EWGs did not lower the barrier of the pericyclic step with the highest barrier calculated for 4-chloro cinnamaldehyde ($\Delta G^{\ddagger} = 12.0$ kcal mol$^{-1}$). Here, exergonicity increases more clearly with donating groups whereas with 4-NO$_2$ it decreases, entry 2, although with 4-Cl substituent, entry 3, it increases slightly. Another validation of the effect of electronic groups on FRS is clarified from distortion/interaction analysis or activation strain model (see below).

## 3.4. Distortion/interaction analysis

Further investigations on the FRS, distortion/interaction analysis have been carried out on the FRS of the cinnamaldehyde and its derivatives (table 2). Estimated contributions of interaction and distortion energies for the FRS from a model for cycloaddition using an approach described by Houk and Bickelhaupt, (for distortion/interaction model see: [67,68]) where activation energy of a reaction can be analysed as a sum of distortion energy and interaction energy. Through this approach, the TS is separated into its components (aldehyde and HOClO) and followed by single-point energy calculations of the obtained separated reactive components. Here, their respective TS geometries must be preserved. Accordingly, the difference in energy between the distorted fragments and optimized ground state geometries represents the distortion energy ($\Delta E_{dis}^{\ddagger,t}$) of the aldehyde and HOClO, whereas the interaction energy is the difference between the activation energy and the distortion energy ($\Delta E_{int}^{\ddagger} = \Delta E^{\ddagger} - \Delta E_{dis}^{\ddagger}$). This model is used to understand the impact of the electronic groups and to realize how proton transfer is a major component of the FRS. Low distortion energy for the aldehyde indicates that most of the activation energy for the rate-determining step is due to the addition of H–[O] moiety into oxygen's carbonyl and not the addition of O = [Cl] moiety into the carbon's

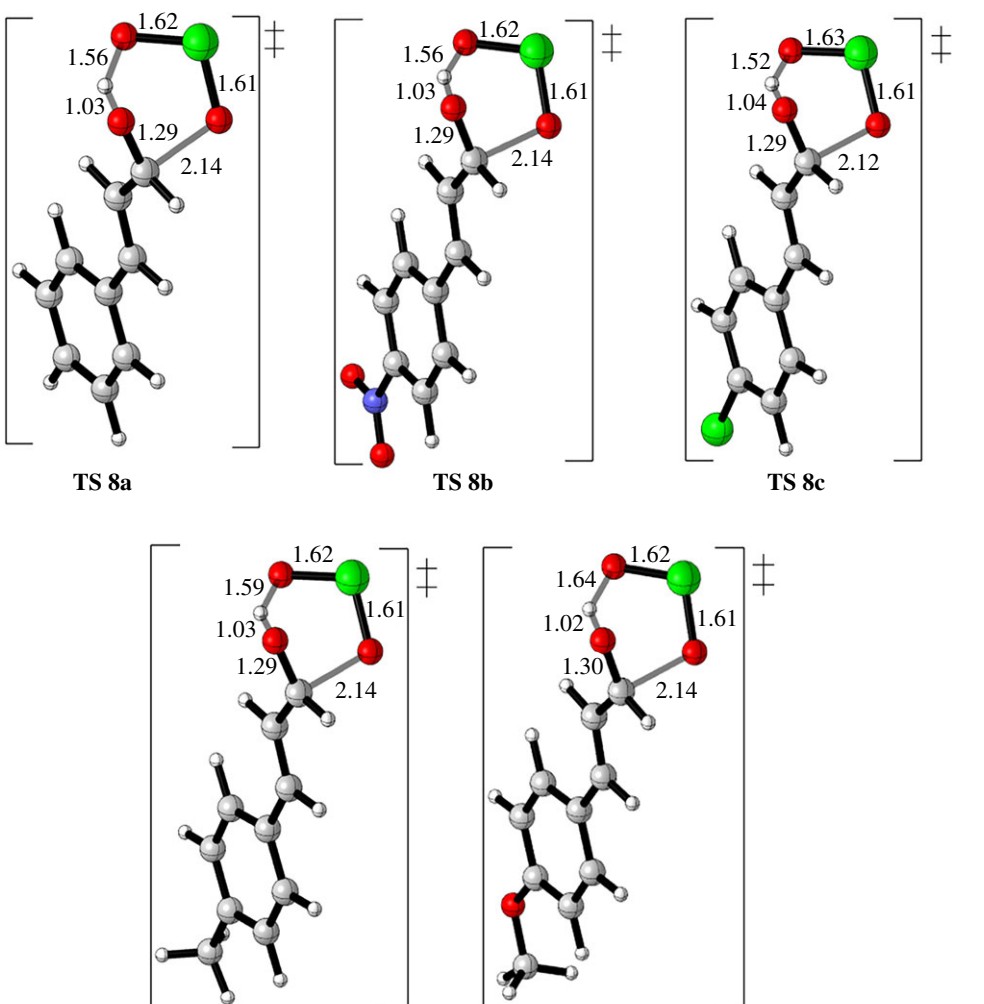

**Figure 9.** FRS TSs of cinnamaldehyde and its derivatives. The TSs and intermediates of the SRS involved in the oxidation of cinnamaldehyde and its derivatives are in the electronic supplementary material.

**Table 2.** Distortion/interaction energies for TSs **8a**–**8e** in kcal mol$^{-1}$. $\Delta E_{dis}^{\ddagger,t}$, $\Delta E_{dis}^{\ddagger,1}$, and $\Delta E_{dis}^{\ddagger,2}$ represents the total, aldehyde and HOClO distortion energies, respectively. $\Delta E_{int}^{\ddagger,t}$ is the interaction energy.

| TS | $\Delta E^{\ddagger}$ | $\Delta E_{dis}^{\ddagger,t}$ | $\Delta E_{dis}^{\ddagger,1}$ | $\Delta E_{dis}^{\ddagger,2}$ | $\Delta E_{int}^{\ddagger,t}$ |
|---|---|---|---|---|---|
| **8a** | 7.3 | 73.4 | 6.9 | 66.5 | −66.1 |
| **8b** | 7.7 | 59.9 | 6.8 | 53.1 | −52.2 |
| **8c** | 7.4 | 68.9 | 7.1 | 61.8 | −61.5 |
| **8d** | 7.0 | 76.2 | 7.1 | 69.6 | −69.2 |
| **8e** | 6.9 | 82.7 | 7.4 | 75.3 | −75.8 |

carbonyl. Very high deformation energy for chlorous acid suggests that the proton of HO–[Cl] moiety is actually closer to the oxygen's carbonyl group, which was supported by IRC and MD simulations, and on the edge of the product zone although the oxygen atom of the O = [Cl] moiety is still in the transition zone and, therefore, needs high energy to distort it. In this manner, EDGs display higher deformation energies with the highest deformation reported with TS **8e** of 82.7 kcal mol$^{-1}$. Also, the highest interaction energies calculated for donating groups are higher than withdrawing groups with the highest interaction found for TS **8e** of −75.8 kcal mol$^{-1}$. Having said that, this interaction represents a major factor in driving the reaction which originally arises from participation of the proton in facilitation of the FRS.

# 4. Conclusion

DFT calculations on Pinnick oxidation of aldehyde into carboxylic acid were carried out using the (SMD)-M06-2X/aug-cc-pVDZ level of theory. Many findings for the reaction were obtained leading to a detailed description of the reaction mechanism that agrees with the experimental observations, importantly the role of acid. It was found that the chlorous acid (HOClO), and in the *cis* conformations, facilitates the FRS through a concerted mechanism, in which a distorted six-membered ring TS was obtained, leading to hydroxyallyl chlorite intermediate with the latter undergoing a pericyclic fragmentation to release the corresponding carboxylic acid in a low barrier ($\Delta G^{\ddagger} = 10$–$12\ \mathrm{kcal\ mol^{-1}}$) and very exergonic step ($\Delta G_r > -100.0\ \mathrm{kcal\ mol^{-1}}$). Results showed that FRS is the rate-determining step. Calculations and experimental $^1$H NMR confirmed that FRS doesn't proceed with the effect of hydrogen bonding from solvent. FMO analysis and IRC and MD calculations gave a fascinating insight about the nature of FRS's TS. Contributions from orbital overlapping between LUMO and HOMO were attributed to the HOClO and aldehyde, respectively, however, the opposite distribution was seen when a strong withdrawing group was present. IRC results showed that most of the calculated barrier of the FRS is evaluated for the proton transfer and not addition of oxo group of O = [Cl] moiety into oxonium ion. A validation on the PES of the FRS was achieved within MD simulation of the FRS; the timing gap between the addition of proton and oxo ligand into carbonyl was found to be 60.0 fs in a dynamically concerted manner. Calculations on cinnamaldehyde and its derivatives highlighted a relatively reduced and increased barrier of the FRS with EDGs and EWGs, respectively. Furthermore, distortion/interaction results added further understanding of the major contributions to the barrier of the FRS, exclusively realizing how high interaction energy arising from acidic facilitation is a major component in driving the FRS. Overall, the present study gives new insight into the reaction mechanism of Pinnick oxidation and indicates the possibility of an inverse-electron demand ene-type reaction during the rate-determining step.

Data accessibility. All the data in this investigation have been reported in the paper and are freely available.

Authors' contributions. A.H. designed the study, coordinated the study, run the simulations and finalized the manuscript; A.A. participated in the writing and discussions; A.M. participated in the data collections; G.M. carried out the molecular laboratory work, participated in data analysis and carried out sequence alignments; F.A. and A.A. participated in the writing and proof reading; S.S. contributed to the writing and proof writing; M.A. conceived of the study. All authors gave final approval for publication.

Competing interests. The authors declare no competing financial interest.

Funding. Unfortunately, this work was done without a financial support.

Acknowledgements. The authors acknowledge the computational resources from the iridis4 supercomputer supported by the University of Southampton. A.A.H. acknowledges the University of Southampton/school of chemistry for providing the visitor-status research position (2717441/EB00-VISIT).

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
