## [Reviewer comments · Royal Society Open Science]

Review History

RSOS-191568.R0 (Original submission)

Review form: Reviewer 1

Is the manuscript scientifically sound in its present form?

No

Are the interpretations and conclusions justified by the results?

No

Is the language acceptable?

No

Do you have any ethical concerns with this paper?

No

Have you any concerns about statistical analyses in this paper?

No

Recommendation?

Reject

Comments to the Author(s)

In their manuscript “*A Dynamically Concerted Transition State in the Oxidation of Aldehydes to Carboxylic Acid by Chlorous Acid*”, Hussein and colleagues computationally analyze the Pinnick oxidation of acrolein derivatives. Despite the exuberant abstract, the paper does not describe a “meticulously high-level density functional theory” analysis. There are so many problems (see below) with this manuscript and I am sorry to say that I currently recommend rejecting the article.

Computational Method:

M06-2X with a double-zeta basis set (aug-cc-pVDZ, not aug-PVDZ as written in the abstract) is NOT a highly accurate and high-level method. I do not want to argue about the problems that M06-2X has compared to other functionals, but a double-zeta basis set typically is associated with a large BSSE. Given the size of the molecules, I would expect a M06-2X/quadruple-zeta basis set to be a “standard” method to calculate electronic in 2019. For a high-level method, I do expect double-hybrid functionals or DLPNO-CCSD(T) calculations with at least triple-zeta basis sets as the minimum.

Furthermore, the authors argue that they employ the SMD solvation model for tBuOH at 298.15 K. As tBuOH has a melting point of 26 °C, the mixture will be a highly viscous solution at best. Therefore, a continuum model will not adequately describe the experimental conditions very well. In fact, as far as I know, tBuOH is usually mixed with other solvents like THF or water. How does this affect the calculations? What is the role of 2-methyl-2-butene that is used as a scavenger in these reactions (in terms of energetic contributions)?

The authors state on page 2 “IRC calculation in the presence of tBuOH was performed”. When inspecting Figure 7, no tBuOH molecule is shown. Do the authors refer to a single explicit tBuOH molecule or the SMD solvation model? This makes a huge difference!

Next, the authors do not comment on the conformational flexibility of their structures. How did the authors analyze the conformational preference for all structures except HOCIO? For that compound, the authors study the conformational preference in the ground state and assume that the same preference is found for all other structures (if I understand the text correctly “we then explored the mechanism of the reaction based on a cis HOCIO”). This might be correct but is not verified here and the Curtin-Hammett principle dictates that this does not have to be the case (i.e., there might be lower energy transition states which adopt a trans HOCIO geometry). This needs to be clarified. In general, I am not sure why this section (HOCIO conformation) needs to be discussed at all in the paper. The conformational preference for HOCIO is known and can be found in typical textbooks for inorganic chemistry.

General Comments

I am missing a Scheme/Figure in the article that summarized the actual experimental conditions that transfer the corresponding aldehydes to carboxylic acids (including solvent, temperature, additives, reaction time, yield, ...). The calculated barrier should then be compared to these conditions.

In Figure 4, the hydrogen-bonded adduct 7t-BuOH shows a cis-orientation. I would have assumed that the alcohol binds from the side of the hydrogen atom (and not the alkyl chain). Have the authors checked this isomeric structure and what is the relative energy.

The discussion of Figure 5 (NMR shifts) is not really clear. Obviously, there is a significant shift upon mixing acrolein and tBuOH. What do the authors mean with the statement “if this slight shift ... is true”? Do they question their own findings here?

Why do the authors study the FMOs in that much detail? If this is considered to be an ene-type reaction, the authors should compare their findings to literature-known examples.

Similarly, the discussion of the IRC is not really clear (or necessary). Obviously, the shape prompted the authors to perform MD simulations. However, these simulations are not really described in the computational section. How many different starting geometries have been selected (e.g., how many different trajectories have been analyzed?). I get the impression that only one trajectory was used. If this is true, the results are statistically irrelevant and should not be discussed. I am surprised to find that the MD simulations did not include implicit tBuOH

molecules. This would give a better answer whether the alcohol plays an important role here or not.

Have the authors included other mechanistic scenarios into their investigations? From the top of my head, I am wondering if an alcohol could act as a proton shuttle through a 7- or 8-membered transition state?

Finally, the authors analyze differently substituted cinnamaldehyde derivatives. Please check the numbering carefully as the number 7 appears frequently with and without letters. Given the accuracy of the chosen method (see above), I am wondering in general how reliable the energetic differences are. How well do they correlate e.g., with Hammett constants? Certainly, it is not necessary to show all structures and transition states on one page. Select 1-2 examples and show and discuss them. Similarly, I am wondering whether the distortion/interaction analysis is meaningful and helpful. The activation energy changed by less than 1 kcal/mol which is clearly within the error margin of the BSSE at the given level.

Minor Comments

The whole manuscript needs a careful editing. I am not a native speaker myself but there are many spelling mistakes, incomplete sentences and otherwise unusual wordings.

Personally, I do not like the wording "first reaction step FRS" as it is a meaningless expression. It might be better to refer to the rate-limiting step which gives additional mechanistic insight.

Why do the authors use numbers in Fig. 1 but not refer to them in the text. This would help to better understand the written statement.

On page 2, right column, the authors write that the proton transfer between dihydrogenphosphate and chlorite is thermodynamically feasible but give a positive reaction free energy. This does not fit.

In the context of the distortion/interaction analysis, the authors should also mention the closely related activation-strain model by Bickelhaupt.

In the Supporting information, please check the numbers for the individual structures (e.g., Structure 8 should be TS8, I guess). Furthermore, please additionally quote the imaginary

Review form: Reviewer 2

Is the manuscript scientifically sound in its present form?

No

Are the interpretations and conclusions justified by the results?

No

Is the language acceptable?

Yes

Do you have any ethical concerns with this paper?

No

Have you any concerns about statistical analyses in this paper?

No

Recommendation?

Major revision is needed (please make suggestions in comments)

Comments to the Author(s)

The paper of Hussein and co-workers presents a computational study on the mechanism of the Pinnick oxidation of aldehydes into carboxylic acids. The paper is focused on the analysis of the

Proton Transfer (PT) and nucleophilic attack (NA) to the carbonyl group of the aldehyde; followed by a second process involving a pericyclic fragmentation. The stepwise and concerted mechanisms were assessed for the PT+NA process; although the work reported only the results for the concerted path, since the TS for the PT of the stepwise mechanism was not found. The authors performed MD and distortion/interaction-activation strain analyses to get further insights on the mechanisms.

The topic of this study is interesting, since Pinnick oxidation is a relevant reaction and this work can provide useful information regarding the reaction mechanism. On this basis, this manuscript can be suitable for publication in Royal Society Open Science. However, some important issues should be addressed:

1. In Figure 2, the depicted stepwise mechanism is incomplete. The TS for the nucleophilic attack to the carbonyl group (after the PT step) should be included in the figure.
2. Even if the TS for the PT step of the stepwise mechanism was not found, the authors should comment about the search for the NA step of the stepwise mechanism. This is important, since the stepwise mechanism cannot be ruled out just because the TS for the PT was not located. Are there available experimental data for the kinetics of this reaction?
3. The MD performed with Gaussian09 correspond to the Born-Oppenheimer Molecular Dynamics (BOMD)? In this regard, the MD of a single trajectory does not provide any meaningful insight in terms of the timescale associated to the mechanism. The MD with quasiclassical (QCT) trajectories should be used instead in order to properly sample the normal modes on the TS before the propagation of the BOMD trajectories (see for example: *Angew. Chem. Int. Ed.* (2014) 53, 8664; and *JACS* (2015) 137, 4749). This normal mode sampling together with the propagation of a few trajectories should be enough to get the correct information regarding the timescale of the involved processes in the reaction mechanism.
4. Distortion-interaction analyses of these types of reactions focusing solely on the TS are not always informative. More insight would likely be obtained by performing distortion-interaction analyses along the reaction coordinate.

Review form: Reviewer 3

Is the manuscript scientifically sound in its present form?

Yes

Are the interpretations and conclusions justified by the results?

Yes

Is the language acceptable?

Yes

Do you have any ethical concerns with this paper?

No

Have you any concerns about statistical analyses in this paper?

No

Recommendation?

Accept with minor revision (please list in comments)

Comments to the Author(s)

The article represents a well designed research work. High-level of DFT calculations on Pinnick oxidation of aldehyde into carboxylic acid were carried out. The theoretical results were validated by the experimental observations. Overall, the present study gives new insight of the reaction mechanism of Pinnick oxidation which might have broad readership. I have only one minor comment for the authors. They should check the whole manuscript for typo errors. There are multiple typo errors throughout the manuscript. One example is the symbol errors especially at page 2, 3 and 5.

Decision letter (RSOS-191568.R0)

19-Nov-2019

Dear Dr Hussein:

Title: A Dynamically Concerted Transition State in the Oxidation of Aldehydes to Carboxylic Acid by Chlorous Acid
Manuscript ID: RSOS-191568

The editor assigned to your manuscript has now received comments from reviewers. I apologise that this has taken longer than usual. We would like you to revise your paper in accordance with the referee and Subject Editor suggestions which can be found below (not including confidential reports to the Editor). Please note this decision does not guarantee eventual acceptance.

Please submit your revised paper before 12-Dec-2019. Please note that the revision deadline will expire at 00.00am on this date. If we do not hear from you within this time then it will be assumed that the paper has been withdrawn. In exceptional circumstances, extensions may be possible if agreed with the Editorial Office in advance. We do not allow multiple rounds of revision so we urge you to make every effort to fully address all of the comments at this stage. If deemed necessary by the Editors, your manuscript will be sent back to one or more of the original reviewers for assessment. If the original reviewers are not available we may invite new reviewers.

Please also include the following statements alongside the other end statements. As we cannot publish your manuscript without these end statements included, if you feel that a given heading is not relevant to your paper, please nevertheless include the heading and explicitly state that it is not relevant to your work.

- Funding statement

Please include a funding section after your main text which lists the source of funding for each author.

RSC Associate Editor:
Comments to the Author:
(There are no comments.)

RSC Subject Editor:
Comments to the Author:
(There are no comments.)

Reviewers' Comments to Author:
Reviewer: 1

Comments to the Author(s)

In their manuscript "*A Dynamically Concerted Transition State in the Oxidation of Aldehydes to Carboxylic Acid by Chlorous Acid*", Hussein and colleagues computationally analyze the Pinnick oxidation of acrolein derivatives. Despite the exuberant abstract, the paper does not describe a "meticulously high-level density functional theory" analysis. There are so many problems (see below) with this manuscript and I am sorry to say that I currently recommend rejecting the article.

Computational Method:

M06-2X with a double-zeta basis set (aug-cc-pVDZ, not aug-PVDZ as written in the abstract) is NOT a highly accurate and high-level method. I do not want to argue about the problems that M06-2X has compared to other functionals, but a double-zeta basis set typically is associated with a large BSSE. Given the size of the molecules, I would expect a M06-2X/quadruple-zeta basis set to be a "standard" method to calculate electronic in 2019. For a high-level method, I do expect double-hybrid functionals or DLPNO-CCSD(T) calculations with at least triple-zeta basis sets as the minimum.

Furthermore, the authors argue that they employ the SMD solvation model for tBuOH at 298.15 K. As tBuOH has a melting point of 26 °C, the mixture will be a highly viscous solution at best. Therefore, a continuum model will not adequately describe the experimental conditions very well. In fact, as far as I know, tBuOH is usually mixed with other solvents like THF or water.

How does this affect the calculations? What is the role of 2-methyl-2-butene that is used as a scavenger in these reactions (in terms of energetic contributions)?

The authors state on page 2 “IRC calculation in the presence of tBuOH was performed”. When inspecting Figure 7, no tBuOH molecule is shown. Do the authors refer to a single explicit tBuOH molecule or the SMD solvation model? This makes a huge difference!

Next, the authors do not comment on the conformational flexibility of their structures. How did the authors analyze the conformational preference for all structures except HOCIO? For that compound, the authors study the conformational preference in the ground state and assume that the same preference is found for all other structures (if I understand the text correctly “we then explored the mechanism of the reaction based on a cis HOCIO”). This might be correct but is not verified here and the Curtin-Hammett principle dictates that this does not have to be the case (i.e., there might be lower energy transition states which adopt a trans HOCIO geometry). This needs to be clarified. In general, I am not sure why this section (HOCIO conformation) needs to be discussed at all in the paper. The conformational preference for HOCIO is known and can be found in typical textbooks for inorganic chemistry.

General Comments

I am missing a Scheme/Figure in the article that summarized the actual experimental conditions that transfer the corresponding aldehydes to carboxylic acids (including solvent, temperature, additives, reaction time, yield, ...). The calculated barrier should then be compared to these conditions.

In Figure 4, the hydrogen-bonded adduct 7t-BuOH shows a cis-orientation. I would have assumed that the alcohol binds from the side of the hydrogen atom (and not the alkyl chain). Have the authors checked this isomeric structure and what is the relative energy.

The discussion of Figure 5 (NMR shifts) is not really clear. Obviously, there is a significant shift upon mixing acrolein and tBuOH. What do the authors mean with the statement “if this slight shift ... is true”? Do they question their own findings here?

Why do the authors study the FMOs in that much detail? If this is considered to be an ene-type reaction, the authors should compare their findings to literature-known examples.

Similarly, the discussion of the IRC is not really clear (or necessary). Obviously, the shape prompted the authors to perform MD simulations. However, these simulations are not really described in the computational section. How many different starting geometries have been selected (e.g., how many different trajectories have been analyzed?). I get the impression that only one trajectory was used. If this is true, the results are statistically irrelevant and should not be discussed. I am surprised to find that the MD simulations did not include implicit tBuOH molecules. This would give a better answer whether the alcohol plays an important role here or not.

Have the authors included other mechanistic scenarios into their investigations? From the top of my head, I am wondering if an alcohol could act as a proton shuttle through a 7- or 8-membered transition state?

Finally, the authors analyze differently substituted cinnamaldehyde derivatives. Please check the numbering carefully as the number 7 appears frequently with and without letters. Given the accuracy of the chosen method (see above), I am wondering in general how reliable the energetic differences are. How well do they correlate e.g., with Hammett constants? Certainly, it is not necessary to show all structures and transition states on one page. Select 1-2 examples and show and discuss them. Similarly, I am wondering whether the distortion/interaction analysis is meaningful and helpful. The activation energy changed by less than 1 kcal/mol which is clearly within the error margin of the BSSE at the given level.

Minor Comments

The whole manuscript needs a careful editing. I am not a native speaker myself but there are many spelling mistakes, incomplete sentences and otherwise unusual wordings.

Personally, I do not like the wording “first reaction step FRS” as it is a meaningless expression. It might be better to refer to the rate-limiting step which gives additional mechanistic insight.

Why do the authors use numbers in Fig. 1 but not refer to them in the text. This would help to better understand the written statement.

On page 2, right column, the authors write that the proton transfer between dihydrogenphosphate and chlorite is thermodynamically feasible but give a positive reaction free energy. This does not fit.

In the context of the distortion/interaction analysis, the authors should also mention the closely related activation-strain model by Bickelhaupt.

In the Supporting information, please check the numbers for the individual structures (e.g., Structure 8 should be TS8, I guess). Furthermore, please additionally quote the imaginary

Reviewer: 2

Comments to the Author(s)

The paper of Hussein and co-workers presents a computational study on the mechanism of the Pinnick oxidation of aldehydes into carboxylic acids. The paper is focused on the analysis of the Proton Transfer (PT) and nucleophilic attack (NA) to the carbonyl group of the aldehyde; followed by a second process involving a pericyclic fragmentation. The stepwise and concerted mechanisms were assessed for the PT+NA process; although the work reported only the results for the concerted path, since the TS for the PT of the stepwise mechanism was not found. The authors performed MD and distortion/interaction-activation strain analyses to get further insights on the mechanisms.

The topic of this study is interesting, since Pinnick oxidation is a relevant reaction and this work can provide useful information regarding the reaction mechanism. On this basis, this manuscript can be suitable for publication in Royal Society Open Science. However, some important issues should be addressed:

1. In Figure 2, the depicted stepwise mechanism is incomplete. The TS for the nucleophilic attack to the carbonyl group (after the PT step) should be included in the figure.
2. Even if the TS for the PT step of the stepwise mechanism was not found, the authors should comment about the search for the NA step of the stepwise mechanism. This is important, since the stepwise mechanism cannot be ruled out just because the TS for the PT was not located. Are there available experimental data for the kinetics of this reaction?
3. The MD performed with Gaussian09 correspond to the Born-Oppenheimer Molecular Dynamics (BOMD)? In this regard, the MD of a single trajectory does not provide any meaningful insight in terms of the timescale associated to the mechanism. The MD with quasiclassical (QCT) trajectories should be used instead in order to properly sample the normal modes on the TS before the propagation of the BOMD trajectories (see for example: *Angew. Chem. Int. Ed.* (2014) 53, 8664; and *JACS* (2015) 137, 4749). This normal mode sampling together with the propagation of a few trajectories should be enough to get the correct information regarding the timescale of the involved processes in the reaction mechanism.
4. Distortion-interaction analyses of these types of reactions focusing solely on the TS are not always informative. More insight would likely be obtained by performing distortion-interaction analyses along the reaction coordinate.

Reviewer: 3

Comments to the Author(s)

The article represents a well designed research work. High-level of DFT calculations on Pinnick oxidation of aldehyde into carboxylic acid were carried out. The theoretical results were validated by the experimental observations. Overall, the present study gives new insight of the reaction mechanism of Pinnick oxidation which might have broad readership. I have only one minor comment for the authors. They should check the whole manuscript for typo errors. There are

multiple typo errors throughout the manuscript. One example is the symbol errors especially at page 2, 3 and 5.

Author's Response to Decision Letter for (RSOS-191568.R0)

See Appendix A.

RSOS-191568.R1 (Revision)

Review form: Reviewer 2

Is the manuscript scientifically sound in its present form?

Yes

Are the interpretations and conclusions justified by the results?

Yes

Is the language acceptable?

Yes

Do you have any ethical concerns with this paper?

No

Have you any concerns about statistical analyses in this paper?

No

Recommendation?

Accept as is

Comments to the Author(s)

This is a revised version of the manuscript titled "Mechanistic Investigations on Pinnick Oxidation: A DFT Study". The authors have addressed the points raised by this reviewer. On this basis, I recommend publication of this work in Royal Society Open Science.

Decision letter (RSOS-191568.R1)

13-Jan-2020

Dear Dr Hussein:

Title: Mechanistic Investigations on Pinnick Oxidation: A DFT Study
Manuscript ID: RSOS-191568.R1

It is a pleasure to accept your manuscript in its current form for publication in Royal Society

Open Science. The chemistry content of Royal Society Open Science is published in collaboration with the Royal Society of Chemistry.

RSC Associate Editor:
Comments to the Author:
I apologise that this has taken longer than usual.

RSC Subject Editor:
Comments to the Author:
(There are no comments.)

Reviewer(s)' Comments to Author:
Reviewer: 2

Comments to the Author(s)
This is a revised version of the manuscript titled "Mechanistic Investigations on Pinnick Oxidation: A DFT Study". The authors have addressed the points raised by this reviewer. On this basis, I recommend publication of this work in Royal Society Open Science.

Appendix A

Dr. Aqeel Alaa Hussein

Dr of Organic and Computational Chemistry.

Visiting Research Position at School of Chemistry, University of Southampton/UK.

Associate Member of Royal Society of Chemistry/UK (AMRSC).

Head of Continuing Education Department at the University of Al-Ameed/Iraq.

Chemistry Lecturer at the University of Al-Ameed/Iraq.

Emails: A.A.H.Hussein@soton.ac.uk, aqeel_alaa85@yahoo.com

25th November 2019

Manuscript: Mechanistic Investigations on Pinnick Oxidation: A DFT Study

Authors: Aqeel A. Hussein,* Azzam A. M. Al-Hadedi, Alaa J. Mahrath, Gamal A. I. Moustafa, Faisal A. Almalki, Alaa Alqahtani, .Sergey Shityakov, Moaed E. Algazally

To the Editor,

We are grateful to the Reviewers and the Editorial Staff for their time and effort to review our manuscript. Please find below our responses to the comments received from the Reviewers and the *Royal Society Open Science* office, which we believe that we have addressed. Any changes have been highlighted in the manuscript. We have changed the title of the manuscript from "A Dynamically Concerted Transition State in the Oxidation of Aldehydes to Carboxylic Acid by Chlorous Acid" to "Mechanistic Investigations on Pinnick Oxidation: A DFT Study" to be more objective.

Sincerely,

Dr. Aqeel A. Hussein

Reviewers' Comments to Author:

Reviewer: 1

In their manuscript “*A Dynamically Concerted Transition State in the Oxidation of Aldehydes to Carboxylic Acid by Chlorous Acid*”, Hussein and colleagues computationally analyze the Pinnick oxidation of acrolein derivatives. Despite the exuberant abstract, the paper does not describe a “meticulously high-level density functional theory” analysis. There are so many problems (see below) with this manuscript and I am sorry to say that I currently recommend rejecting the article.

Computational Method:

We thank the reviewer for his/her valuable suggestions and comments improving the quality of our manuscript.

M06-2X with a double-zeta basis set (aug-cc-pVDZ, not aug-PVDZ as written in the abstract) is NOT a highly accurate and high-level method. I do not want to argue about the problems that M06-2X has compared to other functionals, but a double-zeta basis set typically is associated with a large BSSE. Given the size of the molecules, I would expect a M06-2X/quadruple-zeta basis set to be a “standard” method to calculate electronic in 2019. For a high-level method, I do expect double-hybrid functionals or DLPNO-CCSD(T) calculations with at least triple-zeta basis sets as the minimum.

We totally agree with the reviewer. We have modified the wording that describes our calculation method by deleting the expression “meticulously high-level” from the manuscript.

Furthermore, the authors argue that they employ the SMD solvation model for tBuOH at 298.15 K. As tBuOH has a melting point of 26 °C, the mixture will be a highly viscous solution at best. Therefore, a continuum model will not adequately describe the experimental conditions very well. In fact, as far as I know, tBuOH is usually mixed with other solvents like THF or water. How does this affect the calculations?

As we know that the water would more likely dissolve the NaH₂PO₄ in addition to its roles in reducing the viscosity of the t-BuOH, and we have used t-BuOH because it is a major solvent in the reaction. It is very computationally demanding to include it as an explicit solvent.

What is the role of 2-methyl-2-butene that is used as a scavenger in these reactions (in terms of energetic contributions)?

We have tried to understand the reaction of hypochlorous acid (HOCl) with 2-methyl-2-butene but we couldn't get an appropriate result from our calculations. The literature shows many complex reactions for HOCl in the neutral, acidic and basic conditions which makes the search for transition state modeling

more complicated and that could take the manuscript to more complicated areas which are not the target of the present work.

The authors state on page 2 "IRC calculation in the presence of tBuOH was performed". When inspecting Figure 7, no tBuOH molecule is shown. Do the authors refer to a single explicit tBuOH molecule or the SMD solvation model? This makes a huge difference!

The IRC calculations were performed with SMD solvation model and this was mentioned clearly. Please see:

IRC and MD simulations were carried out with SMD/t-BuOH

Next, the authors do not comment on the conformational flexibility of their structures. How did the authors analyze the conformational preference for all structures except HOClO? For that compound, the authors study the conformational preference in the ground state and assume that the same preference is found for all other structures (if I understand the text correctly "we then explored the mechanism of the reaction based on a *cis* HOClO"). This might be correct but is not verified here and the Curtin-Hammett principle dictates that this does not have to be the case (i.e., there might be lower energy transition states which adopt a *trans* HOClO geometry). This needs to be clarified.

The noticeable structures that need a conformational understanding are HOClO and intermediate **9** and these were studied and shown in this manuscript, and we have mentioned that the TS should take place with the more stable conformer of HOClO which is the *cis* HOClO geometry. Also, if there is a *trans* HOClO geometry during the TS, then we should have a two-step TS (proton transfer and then oxygen addition) but this was not found, and this was also shown in our text:

Within our calculations, a possible TS of proton transfer from HOClO as a separated step from the addition of chlorite ion was not found and, therefore, any complexation between chlorous acid and carbonyl was not formed. Consequently, these results gave the indication that HOClO can be added in a different way, considering different conformations of HOClO.

In general, I am not sure why this section (HOClO conformation) needs to be discussed at all in the paper. The conformational preference for HOClO is known and can be found in typical textbooks for inorganic chemistry.

We think that it is useful to talk about the conformation preference because to our knowledge we couldn't find a text or paper explaining this case, and we wanted to explain why this should occur in this TS geometry.

General Comments

I am missing a Scheme/Figure in the article that summarized the actual experimental conditions that

transfer the corresponding aldehydes to carboxylic acids (including solvent, temperature, additives, reaction time, yield, ...).

We have shown the typical conditions for this reaction and mentioned in Figure 1-a.

The calculated barrier should then be compared to these conditions.

Unfortunately, we couldn't find experimental kinetic data in the literature to compare.

In Figure 4, the hydrogen-bonded adduct **7**_{t-BuOH} shows a cis-orientation. I would have assumed that the alcohol binds from the side of the hydrogen atom (and not the alkyl chain). Have the authors checked this isomeric structure and what is the relative energy.

During optimization, we started with the most comfortable conformation for the **7**_{t-BuOH} and this was the optimized structure. This is in a good agreement with the NMR results. Also, we have tried an optimization based on your suggestions and we got very little energetic difference shown in Fig 4.

The discussion of Figure 5 (NMR shifts) is not really clear. Obviously, there is a significant shift upon mixing acrolein and tBuOH. What do the authors mean with the statement "if this slight shift ... is true"? Do they question their own findings here?

We mean that the reaction still favors to proceed through **TS 8**. Pinnick oxidation can be also conducted in acetonitrile or THF in which such a hydrogen bonding can't be considered. We have also added a sentence to the manuscript to support our opinion:

Moreover, Pinnick oxidation has also been conducted in aprotic solvents like acetonitrile and THF, which adds another evidence to exclude the solvent bonding with **TS 8**.

Why do the authors study the FMOs in that much detail? If this is considered to be an ene-type reaction, the authors should compare their findings to literature-known examples.

Considering this reaction as an ene-type reaction seems sound to us although we found it an inverse-electron demand ene-type one:

This seems to us as an inverse-electron demand ene-type reaction. There is a broad similarity between our findings and those for ene-type reaction.

Similarly, the discussion of the IRC is not really clear (or necessary). Obviously, the shape prompted the authors to perform MD simulations. However, these simulations are not really described in the computational section. How many different starting geometries have been selected (e.g., how many different trajectories have been analyzed?).

The computational procedure is mentioned in the computational details section:

Classical molecular dynamics trajectory calculation on **TS** was initialized in the region of the potential energy surface near to the **TS** and performed using Gaussian 09 in the presence of implicit solvation model (SMD/*t*-BuOH) and under standard conditions ($T = 298.15$ K and 1

atm).¹⁵ The TS was initiated into forward and backward propagations showing the product and reactants (aldehyde and chlorous acid), in which a time step of 1.0 fs was used over periods of 300.0 fs.

And further explanation is mentioned in the results & discussions:

Classical MD calculations were carried out on one trajectory, where the forward and backward propagation of TS's are initiated in the region of the potential energy surface near the TS ($t = 0$ fs) showing the typical reactive bonds toward intermediate **9** and reactants (acrylaldehyde and chlorous acid).

I get the impression that only one trajectory was used. If this is true, the results are statistically irrelevant and should not be discussed.

We have selected one trajectory as we just wanted to see the general dynamically concerted mechanism.

I am surprised to find that the MD simulations did not include implicit *t*-BuOH molecules. This would give a better answer whether the alcohol plays an important role here or not.

The implicit *t*-BuOH through SMD was included in the MD simulations.

Have the authors included other mechanistic scenarios into their investigations? From the top of my head, I am wondering if an alcohol could act as a proton shuttle through a 7- or 8-membered transition state?

Based on our NMR results, and considering the steric hindrance around the OH in *t*-BuOH, the hydrogen bonding is very weak and so we didn't consider a great participation from *t*-BuOH as a proton shuttle in addition to the fact that this suggestion will become very debatable. However, the currently proposed mechanism seems very feasible. As we mentioned above, acetonitrile, for instance, is often employed as solvent in Pinnick oxidation. Such an aprotic solvent can't play a role as proton shuttle.

Finally, the authors analyze differently substituted cinnamaldehyde derivatives. Please check the numbering carefully as the number 7 appears frequently with and without letters.

This has been corrected.

Given the accuracy of the chosen method (see above), I am wondering in general how reliable the energetic differences are. How well do they correlate e.g., with Hammett constants? Certainly, it is not necessary to show all structures and transition states on one page. Select 1-2 examples and show and discuss them.

The level of theory used here shows a very well correlated and reliable result for electronic groups:

DFT calculations show reasonable trends for both EWG and EDG and, generally they correlated very well based on the barriers of TS's **8** and these add further validations to the level of theory being used in this.....

Similarly, I am wondering whether the distortion/interaction analysis is meaningful and helpful. The activation energy changed by less than 1 kcal/mol which is clearly within the error margin of the BSSE at the given level.

It might be that this model is not clearly sound, but it gives us a further validation into the mechanism and effect of electronic groups:

This model is used to understand the reactivity of the electronic groups and to realize how proton transfer is a major component of the FRS.....

Minor Comments

The whole manuscript needs a careful editing. I am not a native speaker myself but there are many spelling mistakes, incomplete sentences and otherwise unusual wordings.

We have checked this and thank you very much for your clarifications.

Personally, I do not like the wording "first reaction step FRS" as it is a meaningless expression. It might be better to refer to the rate-limiting step which gives additional mechanistic insight.

The wording "first reaction step FRS" has been used just for clarity because the mechanism involves two important steps. We have mentioned that the FRS is a rate-limiting step.

Why do the authors use numbers in Fig. 1 but not refer to them in the text. This would help to better understand the written statement.

We have referred to the numbers in Fig 1.

On page 2, right column, the authors write that the proton transfer between dihydrogenphosphate and chlorite is thermodynamically feasible but give a positive reaction free energy. This does not fit.

We have calculated this in H₂O and becomes thermoneutral which is sensible:

the proton-sodium exchange is thermodynamically reasonable ($\Delta G_{r,t-BuOH} = 7.6$ kcal/mol in SMD/t-BuOH or $\Delta G_{r,H_2O} = 0.5$ kcal/mol in SMD/H₂O)

In the context of the distortion/interaction analysis, the authors should also mention the closely related activation-strain model by Bickelhaupt.

It has been added to the text.

In the Supporting information, please check the numbers for the individual structures (e.g., Structure 8 should be TS8, I guess). Furthermore, please additionally quote the imaginary.

This has been corrected and the imaginary frequencies have been added.

Reviewer: 2

Comments to the Author(s)

The paper of Hussein and co-workers presents a computational study on the mechanism of the Pinnick oxidation of aldehydes into carboxylic acids. The paper is focused on the analysis of the Proton Transfer (PT) and nucleophilic attack (NA) to the carbonyl group of the aldehyde; followed by a second process involving a pericyclic fragmentation. The stepwise and concerted mechanisms were assessed for the PT+NA process; although the work reported only the results for the concerted path, since the TS for the PT of the stepwise mechanism was not found. The authors performed MD and distortion/interaction-activation strain analyses to get further insights on the mechanisms.

The topic of this study is interesting, since Pinnick oxidation is a relevant reaction and this work can provide useful information regarding the reaction mechanism. On this basis, this manuscript can be suitable for publication in Royal Society Open Science. However, some important issues should be addressed:

We thank the reviewer for his valuable comments.

1. In Figure 2, the depicted stepwise mechanism is incomplete. The TS for the nucleophilic attack to the carbonyl group (after the PT step) should be included in the figure.

We have corrected this.

2. Even if the TS for the PT step of the stepwise mechanism was not found, the authors should comment about the search for the NA step of the stepwise mechanism. This is important, since the stepwise mechanism cannot be ruled out just because the TS for the PT was not located.

We agree with the importance of the stepwise mechanism, but the issue is that *trans* HOCIO facilitates the concerted mechanism due to the favorable interaction.

Are there available experimental data for the kinetics of this reaction?

Unfortunately, we couldn't find experimental kinetic data in literature, so we have compared the calculated barrier with the typical or overall reaction time, presumably, and found it to be sensible.

3. The MD performed with Gaussian09 correspond to the Born-Oppenheimer Molecular Dynamics (BOMD)? In this regard, the MD of a single trajectory does not provide any meaningful insight in terms of the timescale associated to the mechanism. The MD with quasiclassical (QCT) trajectories should be used instead in order to properly sample the normal modes on the TS before the propagation of the BOMD trajectories (see for example: Angew. Chem. Int. Ed. (2014) 53, 8664; and JACS (2015) 137, 4749). This normal mode sampling together with the propagation of a few trajectories should be enough to get the correct information regarding the timescale of the involved processes in the reaction mechanism.

We have use the ADMP approach rather than BOMD which is recommended by the Gaussian group because it is considerably less computationally demanding and provides equivalent functionality to BOMD:

Classical molecular dynamics trajectory calculation, using the atom-centred density matrix propagation (ADMP),.....

We are not interested in the timescale because we just wanted to see the concertedness of the FRS. Also, we don't have experience in the QCT propagation, and the current approach seems suitable because we just wanted to get further validation about the dynamically concerted mechanism. This would be a future work

4. Distortion-interaction analyses of these types of reactions focusing solely on the TS are not always informative. More insight would likely be obtained by performing distortion-interaction analyses along the reaction coordinate.

It might be that this model is not clearly sound, but it gives us a further validation into the mechanism and effect of electronic groups. Also, it is not always necessary to apply this approach on every single point along the reaction coordinates and we found it in the current way very useful:

This model is used to understand the reactivity of the electronic groups and to realize how proton transfer is a major component of the FRS.....

Reviewer: 3

Comments to the Author(s)

The article represents a well designed research work. High-level of DFT calculations on Pinnick oxidation of aldehyde into carboxylic acid were carried out. The theoretical results were validated by the experimental observations. Overall, the present study gives new insight of the reaction mechanism of Pinnick oxidation which might have broad readership. I have only one minor comment for the authors. They should check the whole manuscript for typo errors. There are multiple typo errors throughout the manuscript. One example is the symbol errors especially at page 2, 3 and 5.

We thank the reviewer for his note. We have corrected these typo errors.